# Assessment of Morpho-Physiological and Biochemical Responses of Mercury-Stressed *Trigonella foenum-gracum* L. to Silver Nanoparticles and *Sphingobacterium ginsenosidiumtans* Applications

**DOI:** 10.3390/plants10071349

**Published:** 2021-07-01

**Authors:** Ahlam Khalofah, Mona Kilany, Hussein Migdadi

**Affiliations:** 1Biology Department, Faculty of Science, King Khalid University, Abha 61413, Saudi Arabia; aalshayeed@kku.edu.sa; 2Research Center for Advanced Materials Science (RCAMS), King Khalid University, Abha 61413, Saudi Arabia; monak@kku.edu.sa; 3Department of Microbiology, National Organization for Drug Control and Research (NODCAR), Giza 12561, Egypt; 4Department of Plant Protection, College of Food and Agriculture Sciences, King Saud University, Riyadh 11461, Saudi Arabia; 5National Agricultural Research Center, Baqa 19381, Jordan

**Keywords:** nanoparticles, mercury, phenolic compounds, fenugreek, pigments, phenol

## Abstract

Heavy metals are primarily generated and deposited in the environment, causing phytotoxicity. This work evaluated fenugreek plants’ morpho-physiological and biochemical responses under mercury stress conditions toward Ag nanoparticles and *Sphingobacterium ginsenosidiumtans* applications. The fabrication of Ag nanoparticles by *Thymus vulgaris* was monitored and described by UV/Vis analysis, FTIR, and SEM. The effect of mercury on vegetative growth was determined by measuring the root and shoots length, the number and area of leaves, the relative water content, and the weight of the green and dried plants; appraisal of photosynthetic pigments, proline, hydrogen peroxide, and total phenols content were also performed. In addition, the manipulation of Ag nanoparticles, *S. ginsenosidiumtans,* and their combination were tested for mercury stress. Here, Ag nanoparticles were formed at 420 nm with a uniform cuboid form and size of 85 nm. Interestingly, the gradual suppression of vegetal growth and photosynthetic pigments by mercury, Ag nanoparticles, and *S. ginsenosidiumtans* were detected; however, carotenoids and anthocyanins were significantly increased. In addition, proline, hydrogen peroxide, and total phenols content were significantly increased because mercury and *S. ginsenosidiumtans* enhance this increase. Ag nanoparticles achieve higher levels by the combination. Thus, *S. ginsenosidiumtans* and Ag nanoparticles could have the plausible ability to relieve and combat mercury’s dangerous effects in fenugreek.

## 1. Introduction

Heavy metals are etiologic agents to all forms of life, even if in tiny amounts. The aggregation of the heavy metals within plant cells is the prime handcuffs on plant growth, which causes plant death and releases heavy metals into the environment by phyto-volatilization [1]. Heavy metals affect the plant’s morphology, physiology, and biochemistry; hence, the typical cell structure, the antioxidant system, and plant growth will be affected and restrict crop production. However, biological means do not effectively destroy heavy metals because of the oxidation state conversion. Some heavy metals may transform into more water-soluble forms that are easily removed by leaching processes, inherently less toxic, precipitate, and change into a specific form that is readily taken away from the impure area. Moreover, photosynthetic pigments are essential parameters in evaluating plant stress and are often used as biomarkers in plants.

A more serious one is mercury, which could be released into the land from various sources such as seed disinfectants, synthetic fertilizers, and herbicides. Mercury is a significant pollutant in toxic materials that has disease registry agency as a priority hazardous substance due to its serious toxic nature, atmospheric motility, and sustained atmospheric residence [2]. The mercury aggregation in plant organs leads to inhibition of plant growth and creates physiological disorders [3]. Mercury adversely affects photosynthesis reactions by exchanging magnesium atoms of chlorophyll, preventing light through the damaged chlorophyll, causing a photosynthesis failure. As *Trigonella foenum-graecum* (fenugreek) is the chief medicinal and edible plant, more studies that grow highly tolerant fenugreek and minimize the mercury accumulation are substantial for food integrity [4]. It was found that plants reposed to heavy metal stresses differently, and alleviating the stress depends on the exposure level of nanoparticles and the different application methods.

Furthermore, the combined application of different kinds of nanoparticles with other materials such as heavy metal-resistant strains of the microbes needs to be investigated. Currently, nanoparticles and beneficial bacteria were employed for improving the plant’s capability against metal resistance. Consistent with nanotechnology’s headway, nanoparticles exhibited either toxic or beneficial plant growth effects [5]. Krishnaraj et al. [6], reported that Ag nanoparticles did not exhibit severe toxic effects. Nowadays, bioremediation employed nanoparticles to remove toxic substances by promoting microbes’ activity [1]. The utmost commercialized investigated nanoparticles are Ag nanoparticles for their various usages and gorgeous physicochemical merits [7]. The enhancement of nanoparticles toward secondary metabolites in medicinal plants is the least studied. Despite nanotechnology being vastly used, its best use in cultivation to promote crop yield is still debated. Discharged ions from industrial activities, fertilizers and agrochemicals, mining, and waste management are exposed to reduction, transforming into nanoparticles, and absorbed by plants, causing industrial impurities that affect the plants’ growth and indirectly affect human health via the food chain. The control of manufacturing effluents in the medium to stop their retrograde effect on vegetation is still considerably low. Thus, it is crucial to investigate their actions on the growth of economic plants such as fenugreek. The biofabrication of Ag nanoparticles is safer and more frugal than others [8]. Notably, the function of microbes in heavy metal biotransformation into benign sorts is well certified. The remedy of the polluted soil with microbes has emerged as the easiest and effective technology by dissolving heavy metals through redox reactions [9]. Various mechanisms such as the binding, volatilizing, immobilizing, oxidizing, and biotransformation of heavy metals enable the microbes to retrieve the media [1]. *Micromonospora* has been recorded as a natural plant endophyte; besides, it boosts aerial parts growth and promotes significant nutrients. Therefore, biofertilizers offer a better choice for stimulating the vegetative growth of horticultural crops in an increasingly eco-mindful world [10]. 

The *S. ginsenosidiumtans* was the selected candidate in the current study because of its worthiness as a bioremediation tool; it enhanced metal removal from aqueous environments [11]. *Sphingobacterium* sp. exploits various agro-residues providing lignocellulolytic enzymes [12], constituting a profitable strategy to develop wholesome rootstock production commercial sweet orange and mandarin plants [13]. Bacterial remediation is extensively used because they grow faster and uptake metals in stress conditions. Moreover, bacteria can adapt harmful metals in the media via accumulating, resisting, adsorbing, and transforming toxic mercury forms to less toxic forms by defense mechanisms [1]. They use different mechanisms, including mercury bioaccumulation, chelation of mercury, sequestration, and blocking mercury entry into cells through permeability barriers and outflow and volatilization to transform harmful ionic mercury, Hg^+2^, to significantly less harmful, elemental mercury, Hg^0^ [14]. 

Therefore, this study aimed to assess the effect of mercury stress on fenugreek growth and assess the morpho-physiological and biochemical responses of mercury-stressed fenugreek to biosynthesized Ag nanoparticles and *S. ginsenosidiumtans* treatments.

## 2. Results

### 2.1. Description of Ag Nanoparticles

The color changed from straw yellow to deep brown because of the excitation of surface plasmon vibrations showing Ag nanoparticles building (Figure 1). The UV-Vis absorption maxima showed a peak near 420 nm, a distinctive peak of Ag nanoparticles, as clarified in Figure 1. FTIR spectra of plant extract (Figure 2) showed a robust broadband between 3000 and 3700 cm^−1^ because of the OH group stretching vibration attributed to alcohols. With SEM analysis help, biofabricated Ag nanoparticles are cuboids approximately 85 nm in size (Figure 3).

### 2.2. Assessment of Plant Growth Traits 

The results showed that the root and shoot lengths were significantly minimized with increasing mercury concentration (2, 4, and 6 mM). In addition, Table 1 shows that all growth traits were significantly affected by increasing mercury concentration (2, 4, 6 mM). Our results showed that Ag nanoparticles and *S. ginsenosidiumtans* minimized the root length, shoot length, leaves number, area of the leaf, fresh weight, and dry weight, and the effect was increased accordingly with elevating HgCl_2_ concentrations.

### 2.3. Photosynthetic Pigments (Chlorophyll a, b, and Total Chlorophyll, Carotenoids), Anthocyanin, and Relative Water Content

As per the present results, the relative water content of fenugreek showed a significant reduction at the highest level (6.0 mM) of HgCl_2_ compared with the non-stressful control. Similarly, the application of Ag nanoparticles and *S. ginsenosidiumtans* and their combination led to a lowering RWC after the second level of HgCl_2_ treatment (Table 2). There was a negative effect of mercury on chlorophyll a, b, and total chlorophyll, especially in higher concentrations (4 and 6 mM); besides, chlorophyll b is sensitive even at low concentrations, as presented in Table 2. In addition, the application of Ag nanoparticles and *S. ginsenosidiumtans* and their combination led to a reduction in chlorophyll pigments. Our findings revealed that carotenoids and anthocyanins pigments rose under all mercury levels. They gradually increased under treatments by *S. ginsenosidiumtans*, Ag nanoparticles, and their combination (Table 2).

### 2.4. Proline, Hydrogen Peroxide (H_2_O_2_) and Total Phenols Content (TPC)

The results (Table 3) showed significant augmentation in proline with increasing mercury concentration (2, 4, and 6 mM) compared to the control. Proline content progressively increased after treating the plant with a combination of *S. ginsenosidiumtans* and Ag nanoparticles. Similarly, H_2_O_2_ content was significantly increased by increasing mercury, and the highest increasing value was at the 6.0 mM HgCl_2_ compared to the control. Over and above, *S. ginsenosidiumtans* and Ag nanoparticles and their combination led to a significant increase in H_2_O_2_ (Table 3). As per the current findings, TPC progressively increased with mercury concentration (2, 4, and 6 mM). Although a non-significant difference was recorded, a successive increase in the TPC was noticed in both stressful and not stressful plants when individually treated with *S. ginsenosidiumtans*, Ag nanoparticles, and their combination as well. However, the most significant difference was observed with the combination of AgNPs and *S. ginsenosidiumtans*.

## 3. Discussion

Fabricated Ag nanoparticles using thyme extract and the change in color from straw yellow to deep brown in this study was because of surface plasmon vibrations and the UV-Vis absorption peak near 420 nm, which is a distinctive peak of Ag nanoparticles [8,15]. The current study provided an eco-friendly, relatively rapid, and cost-saving biogenic protocol to fabricate Ag nanoparticles using thyme extract. Thyme contains reducing agents responsible for lowering silver nitrate to Ag nanoparticles and capping agents inhibiting the accumulation of nanoparticles. The existence of some functional groups of biocompounds was confirmed by FTIR analysis of extract containing silver nanoparticles. FTIR spectra of plant extract showed a robust broadband between 3000 and 3700 cm^−1^ because of the OH group’s stretching vibration attributed to alcohols. Medium bands at 2940 and 2855 cm^−1^ are attributed to the C–H groups assigned to alkanes. The other weak band at 2113 cm^−1^ corresponded to C≡C stretching because of monosubstituted alkynes. A strong peak at 1650 cm^−1^ is assigned to C=C stretching because of monosubstituted alkenes. A strong peak at 1410 and 1254 cm^−1^ may be due to S=O stretching and C-O stretching ascribed to sulfonyl chloride and alkyl aryl ether, respectively. So, the strong band at 1100 cm ^−1^ may be because of C-O-C stretching of ethers [15]. The FTIR spectra of a plant extract containing Ag nanoparticles showed the same plant extract pattern alone but with low intensity, showing that all the compounds mentioned above were consumed in the synthesis procedure [16]. Thyme comprises reducing agents (e.g., flavonoids, phenolic compounds, co-enzymes, and antioxidants such as lutein, zeaxanthin, naringenin, apigenin, thymonin, and luteolin). In addition, the capping agents for Ag nanoparticles of the plant extract, including polysaccharides, lipids, and proteins, limit commercial surfactants [17]. Therefore, the use of biosynthesized Ag nanoparticles represented an alternative way to other ways, which are biocompatible, hydrophilic, and non-toxic. Similarly, the biosynthesis procedures used plant extracts of *Medicago sativa, Pelargonium graveolens, Lemongrass, Azadirachta indica, Cinnamomum Camphor,* and *Aploevera* [18]. 

The phytotoxic effects of mercury on fenugreek growth were assessed by monitoring the root and shoot length, number of leaves and area of the leaf, fresh and dry weight, and relative water content at the different levels of HgCl_2_. Some plants are metal tolerant, where tolerance relies on sequestration, chelation, and exclusion [19]. Heavy metals interfered with the plant metabolic pathways generating signals. However, such an interaction does not indeed have a negatory impact on the plant in all aspects. Antioxidant defense mechanisms minimize the production of reactive oxygen species (ROS); nevertheless, heavy metals cripple the balance between detoxification and ROS generation. Furthermore, heavy metal accumulation altered plant enzymes’ capacities relying on plant species; this clarifies that a robust antioxidant defense strategy helps heavy metal resistance [20]. Therefore, heavy metal concentration and the plant species should be neatly considered when investigating oxidative stress and redox imbalance motivated by heavy metals [19]. 

Our findings showed similarity with Nair and Chung [21]. They showed that Ag nanoparticles significantly reduced root protraction, plant biomass, and fresh weight. Identical results encompass many plants, such as *Brassica nigra* [22], *Lupinus termis* L., and wheat [23,24]. Similarly, the prospect advantage of Ag nanoparticles was investigated, and they caused hazardous interactions with biological systems resulting in phytotoxicity [25]. In addition, Ag nanoparticles have exhibited influences on stimulating the aminocyclopropane-1-carboxylic acid (ACC)-derived inhibition of root extending in Arabidopsis seedlings. In contrast, enhanced seed germination and seedling upgrowth of the *Boswellia* tree have also been investigated upon treatment with Ag nanoparticles [26]. Notably, Ag nanoparticle applications will essentially lean on their physicochemical features (shape, size, surface charge, and solubility). Simultaneously, it was investigated that a clear relationship between the size and toxic relation to the plant where a smaller size was always observed to have higher toxicity to the plant compared to larger Ag nanoparticles [27]. 

The Ag nanoparticles phytotoxicity mechanism is discussed to better understand Ag nanoparticles and plants’ interrelation [28]. Interestingly, there may be a blockage in intercellular communication because of Ag nanoparticles at some points on plasmodesmata and cell walls, affecting nutrient intercellular transport [29]. Concomitantly, Jasim et al. [5], stated that the processing of fenugreek seedlings with Ag nanoparticles appeared to have a significant impact on their vegetative growth parameters. Contrary to our results, it was investigated that the beneficial bacteria significantly promoted leaf length, leaf width, leaf number/plant, and biomass of strawberries [30]. *Sphingobacterium sp.* exploited various agro-residues as a substrate by releasing lignocellulolytic enzymes [12]. Likewise, *Escherichia coli* is five times more mercury-tolerant than *S. aureus* [31]. The co-inoculation of plants with *Sphingobacterium sp.* and *Azotobacter sp.* can represent a successful strategy to develop healthy rootstock to produce commercial mandarin and sweet orange plants [13]. Recent studies have investigated the capacity of microorganisms to perform two-way defense via the production of enzymes for decaying the pollutants and resistance to heavy metals. Several bioremediation mechanisms were recognized, including biosorption, bioaccumulation, biomineralization, microbe–metal interactions, bioleaching, and biotransformation [1].

In this study, the relative water content of fenugreek decreased by increasing the mercury levels compared to the non-stressful control experiment. Moreover, lowering RWC by applying Ag nanoparticles and *S. ginsenosidiumtans* and their combination may be because of the union of mercury to water channel proteins, leading to stomata closure and hindering the water flow in plants [32]. Likewise, it was reported that Cd and Ni^2+^ oxidative stress hinder mustard plant growth and the photosynthesis rate as well as reduce relative water contents. They also notice a significant rise in electrolyte seep, proline contents, and lipid peroxidation [33]. So, RWC is considered an indicator of phytotoxicity when plants are exposed to heavy metal stress [34]. Ag nanoparticles can influence the membrane permeability and, consequently, influence water and nutrient utilization. Ag nanoparticles showed that Ag nanoparticles caused a decrease in the water content of radish sprouts in a dose-dependent manner, showing that Ag nanoparticles might influence plant growth by diminishing water and nutrient content [35]. In contrast these results, *Klebsiella pneumoniae* significantly improved the relative water contents as well as the root and shoot length of wheat plants [36]. 

Photosynthetic pigments are essential parameters in evaluating plant stress and are often used as biomarkers in plants. The result showed a negative effect of mercury on chlorophyll a, b, and total chlorophyll, especially in higher concentrations; besides, chlorophyll b is sensitive even at low concentrations. In addition, the application of Ag nanoparticles and *S. ginsenosidiumtans* and their combination led to a reduction in chlorophyll pigments. Previously, at lower concentrations, the mercury did not substantially affect plant growth, but at higher concentrations, it caused phytotoxicity as well as apparent injuries and physiological defects [32]. Furthermore, several authors have widely reported the reduction of chlorophyll under oxidative conditions [37]. For example, Prasad and Prasad [38] explained that mercury causes the substitution of the magnesium atom in chlorophyll and deactivates the photosynthetic process. Mercury is associated with restraining chlorophyll biosynthesis by joining d-aminolevulinic acid dehydratase [38].

Furthermore, Ag nanoparticle phytotoxicity at the physiological level is because of the mitigation of chlorophyll, carotenoids, and anthocyanins pigments. Ag nanoparticles can hold up the formation of chlorophyll, influencing photosynthesis [39]. Qian et al. [40] showed that Ag nanoparticles were accumulated in Arabidopsis leaves, disrupting the thylakoid membrane structure and decreased chlorophyll level, leading to the prohibition of plant growth. A current study agreed; the results showed that Ag nanoparticles altered the thylakoid in *Physcomitrella* patens, decreased the chlorophyll b level, and imbalanced some central elements in the gametophytes [41]. In *Lupinus termis* L. seedlings, after ten days of continuous exposure to Ag nanoparticles, total protein and total chlorophyll contents were fully diminished [23].

Nevertheless, carotenoids and anthocyanins pigments rose under all mercury levels in this study; they gradually increased under treatments by *S. ginsenosidiumtans*, Ag nanoparticles, and their combination. These findings are corroborated by Thiruvengadam et al. [42]. On the contrary, the exposure to Ag nanoparticles minimized chlorophyll, anthocyanins, and carotenoid contents in *Calendula officinalis* L. [43]. Carotenoids and anthocyanins are non-enzymatic antioxidants safeguarding chlorophyll against ROS on the photosynthetic system by quenching triplet chlorophyll, disrupting chloroplast membrane, and replacing peroxidation [39]. Thence, the generation of these secondary pigments because of metal stresses stimulated the plants’ antioxidant potency to boost the habitual physiological system [44,45]. Previously, it was discovered that Ag nanoparticles speed up oxidative harm in response to ROS release in the plant [46]. Concurrently, carotenoids and phenolics were metabolomics, representing antioxidant compounds in plants released during Ag nanoparticles exposure. The antioxidant defense of carotenoids was attributed to ending the chain reaction of lipoperoxidation in plastids by scavenging active oxygen and barring its formation by bounding chlorophyll [44]. Pan et al. [47] obtained similar results indicating that bacteria could decline oxidative stress by reducing the chlorophyll formation, altering other biochemical and physiological factors because of heavy metals and improving the capacity of plant remediation against heavy metal pollution. *K. ascorbata* was efficient in relieving the growth prohibition caused by heavy metals [48].

Amongst the significant heavy metal stress biomarkers are phenols and proline. Proline is probably associated with metal chelation boosting metal solubility [49]. This study showed a significant augmentation in proline with increasing mercury concentration. The content progressively increased after treating the plant with *S. ginsenosidiumtans* and Ag nanoparticles and their combination. These results align with other literature where proline content was increased when plants faced various stresses where the plant cells have some antioxidants (non-enzymatic) such as proline embroiled in the antioxidant defense responses of plants to Ag nanoparticles, which mitigates the harmful effects of ROS [50]. Elevation of proline levels under stress shows that proline as a cytoplasmic osmolyte prevented protein denaturation [51]. Indeed, Ag nanoparticles are reported to motivate oxidative tension [52]. Simultaneously, another study recorded an increase in proline synthesis under abiotic tension because of beneficial bacteria such as *Burkholderia sp* [53] and *Arthrobacter* and *Bacillus* [54]. Grapevine plants inoculated with *B. phytofirmans* depicted the proline and phenol levels elevation, photosynthesis, and starch deposition [53]. In another respect, the inoculation of the *Helianthus annus* plant with *Lanomicrobium chinense* and *Bacillus cereus* had lowered leaf proline content [55]. Therewithal, the significant proline accumulation in plants under heavy metal stress played a significant role that was summarized in contribution to osmotic modulation at the cellular level, protection from desiccation, protection of enzymes associated with the formula of macromolecules and organelles, relieving the plant stress, and acting as a scavenger of ROS [56].

Herein, H_2_O_2_ content was significantly increased according to increased levels of HgCl_2_ compared to the control, and *S. ginsenosidiumtans*, Ag nanoparticles, and their combination led to a significant increase in H_2_O_2_. Notably, heavy metals produce signs that spur antioxidant defense machinery, lowering H_2_O_2_ levels, although heavy metal tension imbalances the equilibrium between detoxification and ROS generation [20]. The elevation of H_2_O_2_ level in the present study might be attributed to the inactivation of H_2_O_2_ scavenging enzymes. Furthermore, Barba-Espín et al. [57] showed that H_2_O_2_ has stimulated plant growth and development and eased abiotic stresses. The cell membrane has a vital goal of phytotoxicity upon exposition to heavy metal stress, causing membrane perturbation because of hydrogen peroxidation and lipid peroxide, affecting its normal function and texture [34]. Similarly, H_2_O_2_ generation serves as a sign of oxidative stress in some plants such as *Lycopersicon esculentum* and *Brassica juncea* under mercury stress [58]. The principal mechanism underlying Ag nanoparticles phytotoxicity is the excessive ROS production such as H_2_O_2_, resulting in oxidative pressure in plant cells, which is probably because of their nano-sized surface area [39,52]. Furthermore, there are high concentrations of Ag nanoparticles underlying the elevation of ROS production and DNA damage in plant cells [42]. Regardless of the reasons for redox imbalance, many results corroborate oxidative stress participation in response to nanoparticle spraying of beans [59]. Thus, H_2_O_2_ removal may be a remarkable agent in microbial ecology because some members of the microbial communities can release enzymes that are able to detoxify ROS, particularly H_2_O_2_ [60]. Several kinds of bacteria are apparent to be effective scavengers of exogenous H_2_O_2_ [61]. Prior studies involving *Pseudomonas* spp investigated their effectiveness in salinity tolerance by minimizing H_2_O_2_ content [62]. Therefore, further assessment of the influence of beneficial bacteria on the H_2_O_2_ level is essential.

As per the current findings, TPC progressively increased with an increase in mercury. A successive increase in the TPC was noticed in both stressed and not-stressed plants when individually treated with *S. ginsenosidiumtans*, Ag nanoparticles, and their combination. Similarly, mercury-defiled soil exhibited an elevation of phenolics in maize roots [63]. This result is also consistent with that of Soundari et al. [50], who stated that cadmium stimulates tomatoes TPC. El-Beltagi and Mohamed [64] manifested comparable results, too, where an increase in the proline and TPC contents was noticed in the seedlings of *T. foenum-graecum* treated with Pb and Cd. Several plants give off high levels of phenolics when subjected to heavy metals. The de novo biosynthesis of phenolic compounds was boosted under heavy metal stress because of increased enzymes being responsible for phenolic synthesis. Moreover, the main functional hydroxyl and carboxyl groups in the phenolic structure can act as metal chelators and give rise to increase the antioxidant activities and reduce the ROS and lipid alkoxy radical [65]. Thus, phenolics play a vital role in scavenging singlet oxygen and minimizing membrane injuries in the chloroplast. In addition, phenolics are involved in lignin biosynthesis, causing anatomical alteration and resulting in cell wall protection and protecting cells from the harmful action of heavy metals [56]. Similar findings reported that Ag nanoparticles stimulate a higher TPC production [6]. Interestingly, it is evident that the excess of metal and metal derivatives nanoparticles are harmful to plants decreasing TPC, whereas a small amount is beneficial for plants [66]. In agreement with other results, the inoculated plants with *lanomicrobium chinense* and *Bacillus cereus* increased the TPC of *Helianthus annuus* [55].

## 4. Materials and Methods

### 4.1. Reagents and Plants

Chemical substances (AgNO_3_) of pure grade were used (Merck, Ltd., Feltham, UK). The fenugreek seeds (*Trigonella foenum graceum* L.) used in this study were provided by the Ministry of Agriculture, Abha (Saudi Arabia). Common thyme *(Thymus vulgaris)* fresh leaves were obtained from the local market of Abha.

### 4.2. Bacterial Inoculant Preparation

The *S. ginsenosidiumtans* were cultured from the soil rhizosphere, Aseer region, Saudi Arabia. Bacterial culture was grown overnight on nutrient agar at 150 rpm and 27 °C to give an OD = 0.1 at 600. Bacterial supernatant was prepared by centrifugation of the culture for 15 min at 5000 rpm. All pots were drizzled with bacterial supernatant for foliar application, and the control pots were drizzled with a similar volume of disinfected water [67].

### 4.3. Thymus Vulgaris-Mediated Ag Nanoparticles Biosynthesis

The mature leaves of *Thymus vulgaris* were rinsed using distilled water, shade-dried at 20–25 °C for five days, and then ground. About 10 g of ground leaves were soaked in 100 mL of distilled water at 25 °C for 24 h; then, they were filtered and finally centrifuged for 10 min at 5000 rpm to separate the clear aquatic leaf extract that was preserved at 4 °C until its use as a reducing and stabilizer agent in nanoparticles preparation [8]. 

### 4.4. Description of Ag Nanoparticles 

The initial indicator for Ag nanoparticles formation is the change in color through 24 h. The characterization of Ag nanoparticles was examined via UV-Vis spectroscopy analysis using a UV-3600 Shimadzu spectrophotometer and monitored within the range 200–600 nm. Fourier transform infrared spectroscopy (FTIR) was done using Perkin Elmer Spectrum 2000, USA, at a rate of 16 times within the range 600–4000 cm^−1^ and clarity of 4 cm^−1^. A scanning electron microscope explored the physical characteristics of the produced Ag nanoparticles (shape and size) at a quickening voltage of 90 kV (SEM, JEM-1011, JEOL, Tokyo, Japan).

### 4.5. Phenotypic Assay

Before planting, fenugreek seeds were exterior-sterilized by vortexing for 5 min in 70% ethyl alcohol and 2% sodium hypochlorite for 30 min. Afterward, aseptic seeds were washed for 5 min with disinfected water. Then, exterior-sterilized seeds were planted in 15 cm perforated plastic pots containing sand and peat moss (1:1 volumes). Fifty seeds were evenly distributed at approximately 1 cm deep in each pot. HgCl_2_ treatments were 0, 2, 4, and 6 mM. The pots were kept in the greenhouse at 20–25 °C and exposed to daylight. First, pots were irrigated thrice a week with 200 mL of water; then, the pots were divided into four groups; each group contains three replicates. The first group is the control (0.0 mM HgCl_2_) using water and included treating plants with Ag nanoparticles, the bacterial supernatant of *S. ginsenosidiumtans*, and a mixture of Ag nanoparticles and *S. ginsenosidiumtans* bacteria supernatant. Similarly, Ag nanoparticles and bacterial treatments were applied for HgCl_2_-treated pots, wherein the 2.0 mM HgCl_2_ treatment was the second group, while 4.0 mM and 6.0 mM were the third and fourth groups, respectively. Then, 200 mL of 2, 4, or 6 mM HgCl_2_ every week per pot was applied for three weeks. Hence, there were sixteen treatments. The treatment lasted for 21 days; after that, one week later, the shoots and roots were separated and washed before the vegetative and physiological examination. The fenugreek’s vegetal growth was determined by the recording root length, shoot length, leaves number, and leaf area.

### 4.6. Determination of Fresh Weight, Dry Weight, and Relative Water Content 

For biomass, carefully uprooted plants were thoroughly washed with distilled water to remove adhered sand particles. Following the gathering of plants, fresh weight (FW) was instantly gauged, and plants were oven-dried at 65 °C for 72 h for the dry weight (DW) determination [58]. Dependently, relative water content (RWC) was estimated following the next equation:RWC (%) = [(FW − DW)/FW] × 100.(1)

### 4.7. Assessment of the Photosynthetic Pigments (Chlorophyll a, b, and Total Chlorophyll, and Carotenoids)

About 0.2 g green fresh leaves from control and treated seedlings were ground in 10 mL of 80% (*v*/*v*) ice-cooled acetone in dark conditions, and chlorophyll contents were fully extracted by 100% acetone and measured at wavelengths of 647 and 663 nm for chlorophyll assays [68]. The formula described by Afroz. et al. [69] was adopted for colorimetric determination of the carotenoid content at 470 nm. 

### 4.8. Assessment of H_2_O_2_, Proline, Total Phenols Contents (TPC), and Anthocyanins

The H_2_O_2_ level was measured colorimetrically [70]. An aliquot of 200 μL acetone extract was mixed with 0.04 mL of 0.1% TiO_2_ and 0.2 mL of NH_4_OH (20%). The pellet was decollated with acetone and resuspended in 0.8 mL of H_2_SO_4_. Then, the mixture was centrifuged at 6000 g for 15 min, and the supernatant was read at 415 nm. For proline determination, 0.5 g of dry leaves was homogenized with 5 mL of 3% sulfosalicylic acid. The reaction mixture containing 2 mL of 1% of Ninhydrin (*w*/*v*) in 60% glacial acetic acid (*v*/*v*) and 20% ethanol (*v*/*v*) was collectively boiled at 100 °C for 30 min, followed by extraction using 6 mL of toluene after cooling. Thoroughly mixing resulted in the chromophore’s separation, which was detected at 520 nm [71]. The standard proline curve (mg/g weight) was used to estimate the amount of proline in the plant sample. TPC was determined by mixing 0.75 mL Folin–Ciocalteu reagent with 100 µL plant extract and incubated at 22 °C for 5 min. Then, 0.75 mL Na_2_CO_3_ solution was combined with the previous mixture and kept at 22 °C for 90 min. TPC was monitored at 725 nm with a UV/Vis-DAD spectrophotometer [72]. Gallic acid (GA) was used as a reference standard, and TPC was estimated from the GA calibration curve (range 5–200 µg/mL). Frosted tissues were steeped instantly in acidified methyl alcohol (methanol, water, HCl: 16, 3, 1) and then mashed and kept for 72 h at 25 °C in a dark place. The proportional content of anthocyanin was spectrophotometrically estimated at 530 nm and 653 nm [73].

### 4.9. Statistical Analyses

In the experiment, 50 plants were randomly attributed to each treatment and replicated three times. The data were analyzed by one-way ANOVA and the honestly significant difference (HSD) at *p* < 0.05 probability level using Tukey post hoc test used to compare the differences among treatment means using SAS software (version 9.1 Institute, Cary, NC, USA).

## 5. Conclusions

The current results showed that the application of Ag nanoparticles and *S. ginsenosidiumtans* could increase adverse mercury effects on the *Trigonella foenumgracum* plant by reducing vegetative growth RWC and chlorophyll contents. Furthermore, safeguarding the plant upon mercury stress was achieved via increasing the carotenoid contents, anthocyanins, proline, H_2_O_2,_ and total phenols content enhanced by Ag nanoparticles in *S. ginsenosidiumtans*. Thus, the application of Ag nanoparticles and *S. ginsenosidiumtans* could help to energize the growth and economic yield in plants growing stressed by heavy metals. However, to understand how Ag nanoparticles and *S. ginsenosidiumtans* mitigate the pernicious outcomes of heavy metal stress in plants, further efforts are required. Another essential additive component of the plant defense system is a symbiotic association with silver nanoparticles and rhizosphere bacteria that can effectively immobilize mercury and reduce its uptake by plants. Additionally, they can enhance the activities of the antioxidant defense machinery of plants.

## Figures and Tables

**Figure 1 plants-10-01349-f001:**
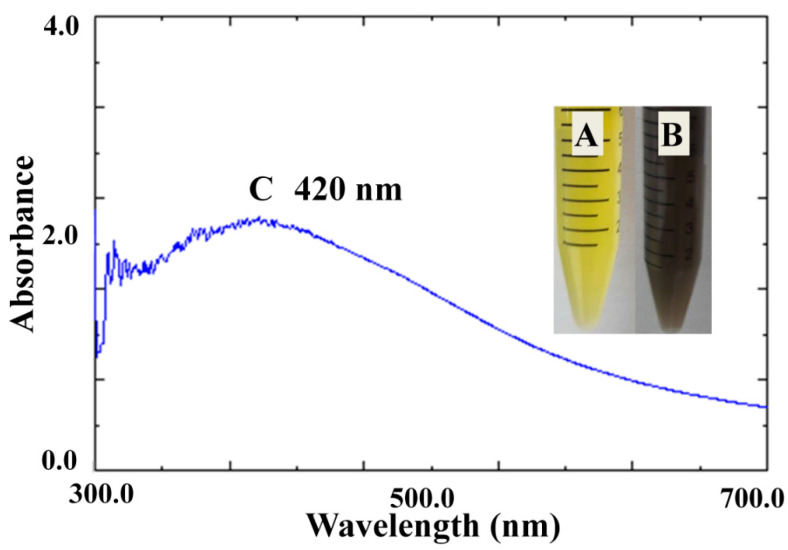
Color change and UV-Visible spectral analysis of plant extract with Ag nanoparticles. Where (**A**) denotes plant extract, (**B**) denotes plant extract and AgNO_3_, and (**C**) denotes spectral analysis of synthesized Ag nanoparticles.

**Figure 2 plants-10-01349-f002:**
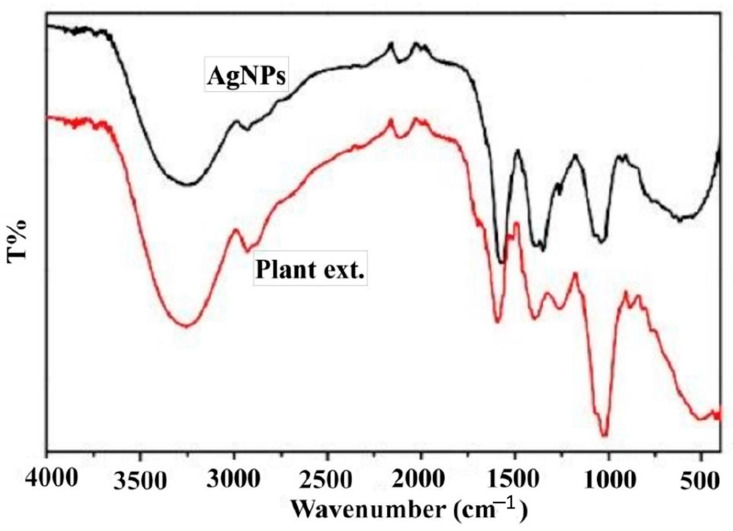
FTIR spectrum of biosynthesized Ag nanoparticles.

**Figure 3 plants-10-01349-f003:**
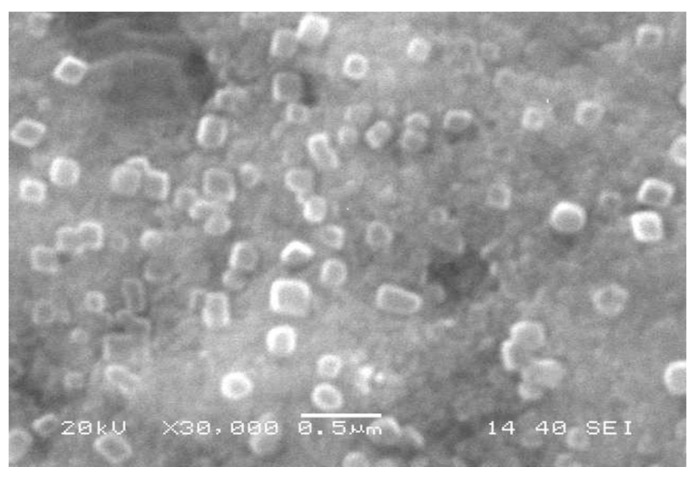
FTIR spectrum of biosynthesized Ag nanoparticles.

**Table 1 plants-10-01349-t001:** Effect of AgNPs, *Sphingobacterium ginsenosidiumtans* (S. g), and their combination on *Trigonella foenum-gracum* L. plant growth parameters under HgCl_2_ stress (0, 2, 4, and 6 mM).

Treatments	Root Length (cm)	Shoot Length (cm)	Number of Leaves/Plant	Area of Leaf (cm^2^)	Fresh Weight (mg)	Dry Weight (mg)
0.0 mM HgCl_2_	9.03 ± 0.52 a	15.29 ± 0.99 a	5.78 ± 0.04 a	3.37 ± 0.02 a	33.89 ± 0.19 a	12.59 ± 0.93 a
AgNPs	8.98 ± 0.79 a	14.76 ± 0.48 ab	5.63 ± 0.03 a	3.24 ± 0.01 b	33.67 ± 0.16 ab	11.37 ± 0.82 b
S. g	8.16 ± 0.48 b	14.53 ± 0.74 b	5.61 ± 0.07 a	3.19 ± 0.03 bc	32.30 ± 0.14 bc	11.12 ± 0.21 b
AgNPs + S. g	8.13 ± 0.14 b	14.42 ± 0.14 bc	5.61 ± 0.03 a	3.17 ± 0.04 bc	32.45 ± 0.12 bc	11.04 ± 0.34 b
2.0 mM HgCl_2_	7.32 ± 0.90 c	13.38 ± 0.88 cd	5.06 ± 0.02 b	2.97 ± 0.03 cd	30.08 ± 0.12 cd	8.45 ± 0.76 c
2.0 mM HgCl_2_ + AgNPs	7.13 ± 0.06 c	13.24 ± 0.12 d	4.98 ± 0.06 b	2.84 ± 0.03 de	29.36 ± 0.23 de	8.37 ± 0.86 c
2.0 mM HgCl_2_ + S. g	7.06 ± 0.23 c	13.11 ± 0.27 d	4.90 ± 0.05 b	2.79 ± 0.09 e	29.11 ± 0.15 df	8.31 ± 0.29 c
2.0 mM HgCl_2_ + AgNPs + S. g	7.04 ± 0.55 c	12.89 ± 0.58 d	4.86 ± 0.09 b	2.75 ± 0.01 e	29.02 ± 0.19 df	8.28 ± 0.42 c
4.0 mM HgCl_2_	6.26 ± 0.88 d	9.28 ± 0.19 e	3.99 ± 0.08 c	1.17 ± 0.04 f	26.88 ± 0.18 eg	5.41 ± 0.83 d
4.0 mM HgCl_2_ + AgNPs	6.01 ± 0.35 de	9.13 ± 0.33 e	3.81 ± 0.07 cd	1.04 ± 0.00 fg	26.43 ± 0.17 fg	5.24 ± 0.93 d
4.0 mM HgCl_2_ +S. g	5.81 ± 0.29 e	9.07 ± 0.27 e	3.78 ± 0.04 cd	0.99 ± 0.03 g	26.21 ± 0.22 gh	5.19 ± 0.64 d
4.0 mM HgCl_2_ + AgNPs + S. g	5.66 ± 0.69 e	8.99 ± 0.16 e	3.73 ± 0.02 d	0.98 ± 0.03 g	26.19 ± 0.14 gh	5.16 ± 0.32 d
6.0 mM HgCl_2_	3.35 ± 0.78 f	5.76 ± 0.74 f	3.01 ± 0.03 e	0.93 ± 0.02 g	24.06 ± 0.12 hi	3.11 ± 0.54 e
6.0 mM HgCl_2_ + AgNPs	3.23 ± 0.55 f	5.47 ± 0.56 f	2.74 ± 0.05 ef	0.88 ± 0.07 gh	23.22 ± 0.16 i	3.02 ± 0.83 ef
6.0 mM HgCl_2_ +S. g	3.10 ± 0.33 f	5.31 ± 0.38 f	2.66 ± 0.08 f	0.76 ± 0.01 h	23.19 ± 0.19 i	2.74 ± 0.74 ef
6.0 mM HgCl_2_ + AgNPs+ S. g	3.02 ± 0.71 f	5.26 ± 0.79 f	2.59 ± 0.04 f	0.74 ± 0.05 h	23.04 ± 0.22 i	2.58 ± 0.54 f
An honestly significant difference (HSD) at *p* < 0.05 probability level using Tukey’s test						
	0.48	0.85	0.33	0.17	2.14	0.58

Mean ± SD values for treatment over three replications. According to Tukey’s test, different letters within the same columns show significant differences (*p* < 0.05).

**Table 2 plants-10-01349-t002:** The effect of AgNPs, *Sphingobacterium ginsenosidiumtans* (S. g), and their combination on relative water content (RWC), chlorophyll a, chlorophyll b, total chlorophyll, carotenoids, and anthocyanins pigments of *Trigonella foenum-gracum* L. plants under HgCl_2_ (0, 2, 4, and 6 mM).

Treatments	RWC (%)	Chllorophyll a (mg/g)	Chllorophyll b (mg/g)	Total Chlorophyll (mg/g)	Carotenoids (mg/g)	Anthocyanins (mg/g)
0.0 mM HgCl_2_ AgNPs S. g AgNPs + S. g	98.01 ± 1.24 a	2.01 ± 0.06 a	0.98 ± 0.03 a	2.99 ± 0.13 a	0.73 ± 0.03 h	0.76 ± 0.09 h
AgNPs	97.44 ± 1.78 ab	1.91 ± 0.02 b	0.95 ± 0.05 ab	2.86 ± 0.09 ab	0.77 ± 0.05 h	0.79 ± 0.08 gh
S. g	97.23 ± 1.63 a–c	1.85 ± 0.01 b	0.93 ± 0.07 bc	2.78 ± 0.46 bc	0.75 ± 0.04 h	0.77 ± 0.05 h
AgNPs + S. g	97.04 ± 1.54 a–d	1.82 ± 0.05 bc	0.89 ± 0.04 c	2.71 ± 0.08 c	0.81 ± 0.02 gh	0.81 ± 0.02 f–h
2.0 mM HgCl_2_ 2.0 mM HgCl_2_ + AgNPs 2.0 mM HgCl_2_ + S. g 2.0 mM HgCl_2_ + (AgNPs + S. g)	95.89 ± 1.90 a–d	1.79 ± 0.09 bc	0.79 ± 0.08 d	2.58 ± 0.17 cd	0.79 ± 0.03 gh	0.82 ± 0.03 e–h
AgNPs	95.46 ± 1.69 a–d	1.70 ± 0.03 cd	0.75 ± 0.09 de	2.45 ± 0.15 de	0.84 ± 0.01 e–g	0.86 ± 0.08 d–f
S. g	95.37 ± 1.23 a–e	1.68 ± 0.03 d	0.74 ± 0.09 ef	2.42 ± 0.07 ef	0.81 ± 0.02 f–h	0.83 ± 0.03 e–h
AgNPs + S. g	95.32 ± 1.90 a–e	1.63 ± 0.10 d	0.73 ± 0.04 ef	2.36 ± 0.19 ef	0.85 ± 0.07 e–h	0.85 ± 0.02 e–g
4.0 mM HgCl_2_ 4.0 mM HgCl_2_ + AgNPs 4.0 mM HgCl_2_ + S. g 4.0 mM HgCl_2_ + (AgNPs + S. g)	91.24 ± 1.07 a–f	1.54 ± 0.06 de	0.68 ± 0.02 fg	2.22 ± 0.63 fg	0.86 ± 0.05 d–f	0.87 ± 0.09 c–e
AgNPs	91.01 ± 1.58 b–g	1.49 ± 0.08 ef	0.64 ± 0.08 gh	2.13 ± 0.05 gh	0.89 ± 0.01 c–e	0.92 ± 0.06 a–d
S. g	90.71 ± 1.27 c–g	1.45 ± 0.01 fg	0.62 ± 0.04 hi	2.07 ± 0.93 hi	0.87 ± 0.06 e–g	0.89 ± 0.01 b–e
AgNPs + S. g	90.65 ± 1.65 d–g	1.42 ± 0.02 fg	0.60 ± 0.02 hi	2.02 ± 0.04 hi	0.91 ± 0.09 b–e	0.96 ± 0.08 ab
6.0 mM HgCl_2_	87.17 ± 1.89 e–g	1.46 ± 0.04 f	0.56 ± 0.07 ij	2.02 ± 0.16 h–j	0.93 ± 0.06 a–c	0.93 ± 0.03 a–c
6.0 mM HgCl_2_ + AgNPs	86.93 ± 1.54 f–g	1.38 ± 0.09 f–h	0.53 ± 0.09 jk	1.91 ± 0.03 i–k	0.96 ± 0.01 ab	0.97 ± 0.04 a
6.0 mM HgCl_2_ + S. g	86.10 ± 1.22 f–g	1.34 ± 0.06 gh	0.51 ± 0.08 k	1.85 ± 0.15 jk	0.94 ± 0.07 a–d	0.95 ± 0.04 a–c
6.0 mM HgCl_2_ + AgNPs + S. g	86.01 ± 1.22 g	1.31 ± 0.05 h	0.48 ± 0.06 k	1.79 ± 0.02 k	0.98 ± 0.04 a	0.99 ± 0.09 a
An honestly significant difference (HSD) at *p* < 0.05 probability level using Tukey’s test for:						
HgCl_2_ treatments	7.10	0.12	0.05	0.17	0.07	0.07

Mean ± SD values for treatment over three replications. According to Tukey’s test, different letters within the same columns show significant differences (*p* < 0.05).

**Table 3 plants-10-01349-t003:** Effect of AgNPs, *Sphingobacterium ginsenosidiumtans* (S. g), and their combination on proline, hydrogen peroxide, and total phenols content of *Trigonella foenum-gracum* L. plants under HgCl_2_ stress (0, 2, 4, and 6 mM).

Treatments	Proline (mg/g)	Hydrogen Peroxide (µmol/g)	Total Phenols Content (mg/g)
0.0 mM HgCl_2_ AgNPs S. g AgNPs + S. g AgNPs S. g AgNPs + S. g	1.35 ± 0.03 f	0.21 ± 0.04 i	1.28 ± 0.63 f
	1.38 ± 0.06 aef	0.24 ± 0.06 gh	1.35 ± 0.44 f
	1.37 ± 0.03 f	0.23 ± 0.08 hi	1.32 ± 0.04 f
	1.41 ± 0.01 d–f	0.26 ± 0.10 fg	1.39 ± 0.50 f
2.0 mM HgCl_2_ 2.0 mM HgCl_2_ + AgNPs 2.0 mM HgCl_2_ + S. g 2.0 mM HgCl_2_ + (AgNPs + S. g) AgNPs S. g AgNPs + S. g	1.43 ± 0.09 b–f	0.23 ± 0.04 hi	1.54 ± 0.95 de
	1.47 ± 0.08 a–f	0.28 ± 0.02 ef	1.59 ± 0.62 c–e
	1.45 ± 0.05 c–f	0.27 ± 0.01 f	1.56 ± 0.38 e
	1.51 ± 0.02 a–d	0.31 ± 0.04 d	1.62 ± 0.27 b–e
4.0 mM HgCl_2_ 4.0 mM HgCl_2_ + AgNPs 4.0 mM HgCl_2_ + S. g 4.0 mM HgCl_2_ + (AgNPs + S. g) AgNPs S. g AgNPs + S. g	1.48 ± 0.01 a–d	0.29 ± 0.01 de	1.66 ± 0.34 b–e
	1.52 ± 0.09 a–c	0.35 ± 0.06 c	1.71 ± 0.96 bc
	1.50 ± 0.02 a–e	0.31 ± 0.03 d	1.69 ± 0.54 b–d
	1.56 ± 0.08 a–c	0.38 ± 0.03 b	1.74 ± 0.47 b
6.0 mM HgCl_2_	1.53 ± 0.04 a–c	0.31 ± 0.02 d	1.87 ± 0.35 a
6.0 mM HgCl_2_ + AgNPs	1.56 ± 0.04 ab	0.37 ± 0.04 bc	1.91 ± 0.18 a
6.0 mM HgCl_2_ + S. g	1.54 ± 0.03 a–c	0.35 ± 0.06 c	1.89 ± 0.84 a
6.0 mM HgCl_2_ + AgNPs + S. g	1.58 ± 0.02 a	0.41 ± 0.03 a	1.93 ± 0.69 a
An honestly significant difference (HSD) at *p* < 0.05 probability level using Tukey’s test for:			
treatments	0.11	0.02	0.13

Mean ± SD values for treatment over three replications. According to Tukey’s test, different letters within the same columns show significant differences (*p* < 0.05).

## Data Availability

The data presented in this study are available on request from the corresponding author.

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
