# Peer review of "Assessment of Morpho-Physiological and Biochemical Responses of Mercury-Stressed Trigonella foenum-gracum L. to Silver Nanoparticles and Sphingobacterium ginsenosidiumtans Applications"

_plants, 2021, doi:10.3390/plants10071349_

Round 1

Reviewer 1 Report

In the present manuscript, Authors evaluated the effect of Ag-nanoparticles and soil microbes’ application on Hg-treated fenugreek plants. Although the topic seems to be interesting, the novelty of this work should be more clearly highlighted. The Introduction does not provide sufficient information for what the study was conducted and why such physiological parameters were determined. Such statements can be found in the discussion section, whose purpose is to interpret and describe the significance of obtained findings in light of what was already known about the research problem rather than to justify why such and not other issues were taken into consideration. What is more, the discussion is definitely too long and it consists of unrelated to each other paragraphs resulting in the descriptive presentation of obtained findings instead of the explanation of general mechanisms functioning in stressed plants. Too much attention is paid to the technology of NPs production as well as to the impact of Ag-NPs, although the purpose of the study was different. It should be also noticed that the application of Ag-NPs within particular Hg treatments did not change anything in tested parameters. Therefore, the discussion cannot be based on AgNPs impacts, because they were not observed in the current study! Please, remove also any statements which do not concern the present experiments (e.g. about bioremediation or PSII, etc., unless you can provide data about heavy metal concentration in the soil and plant tissues or photosynthetic efficiency, respectively. Undoubtedly, it could improve the quality of the manuscript). To sum up, the discussion should be significantly remodeled in respect to the aim of undertaken study and to show multidimensional and interrelated scientific truth resulting from conducted research. I also encourage the Authors to learn more about oxidative stress and their impact not only on chloroplasts.

Conclusions are written exaggeratedly and they are not supported by the results obtained. Moreover, they contradict each other when the statement that ‘the application of Ag-nanoparticles and S. ginsenosidiumtans increased adverse mercury effects on the Trigonella foenumgracum plant by reducing vegetative growth RWC and chlorophyll contents’ is taken into account together with the following one about AgNPs and S.g. usefulness ‘to energize the growth and yield in stressed plants’. The increased level of H2O2 indicated oxidative stress and it is not absolutely connected with stress mitigation! An enhanced accumulation of mentioned non-enzymatic antioxidants together with H2O2 content seems to confirm this hypothesis. It cannot be also concluded about mercury uptake since its content was not determined. Undoubtedly, Ag-NPs or/and symbiotic microorganisms may help to combat stress conditions, but it was not observed in presented results.

Some detailed comments:

Key words: delate thyme; phenolic compounds would be better

Statements in lines 79-80, 102-104, 119-120, etc. are not the results. Please, avoid any general comments in the result section and focus only on obtained data.

Line 104: these features were monitored not only at higher concentration. By the way, higher than what?

The abbreviation ‘ns’ does not occur in the tables.

Line 117: what does the ‘second level’ mean?

Line 139: 6 instead of six

Table 2. complete units with FW (fresh weight) or DW (dry weight)

Please, remove brackets when NPs and S.g. combination is mentioned in every tables.    

Line 359: provide full name and explain how was this species identified and obtained.

How was the HgCl2 applied? Once? How were the plants watered during cultivation? How was the leaching of Hg from the soil stopped?

Line 411: change the order on H2O2, proline, TPC (it refers also to the description)

Line 426: Reference [74] is redundant; by the way, it does not exist in the reference list.

Anthocyanins are not the photosyntetic pigments! They represent phenolic compounds and the procedure for their extraction/determination should be described together with TPC.

Specify for which analysis the ninhydrin was used. Explain how the proline and TPC levels were estimated (using calibration curve?) Equivalent of which reagents were used for quantification?. Provide more detailed information about concentrations of used reagents for H2O2 determination.  

The unit fo H2O2 is unacceptable and values must be recalculated in a proper way for this parameter in order to show the H2O2 content per gram (not l) of tissue.

There are numerous typing and language errors, that could be carefully removed from the whole text.

Author Response

Dear professor

We want to express our thanks to you for your valuable suggestion and comments, which help us improve our manuscript. With pleasure, all comments are considered and inserted in the main manuscript.

  • In the present manuscript, Authors evaluated the effect of Ag-nanoparticles and soil microbes’ application on Hg-treated fenugreek plants. Although the topic seems to be interesting, the novelty of this work should be more clearly highlighted.
  • The Introduction does not provide sufficient information for what the study was conducted and why such physiological parameters were determined. Such statements can be found in the discussion section, whose purpose is to interpret and describe the significance of obtained findings in light of what was already known about the research problem rather than to justify why such and not other issues were taken into consideration.

Thank you very much. Some information was added to the introduction section.

  • What is more, the discussion is definitely too long and it consists of unrelated to each other paragraphs resulting in the descriptive presentation of obtained findings instead of the explanation of general mechanisms functioning in stressed plants.
  • Thank you very much. Some were added to the introduction section.

  • Too much attention is paid to the technology of NPs production as well as to the impact of Ag-NPs, although the purpose of the study was different. It should be also noticed that the application of Ag-NPs within particular Hg treatments did not change anything in tested parameters. Therefore, the discussion cannot be based on AgNPs impacts, because they were not observed in the current study! Please, remove also any statements which do not concern the present experiments (e.g. about bioremediation or PSII, etc., unless you can provide data about heavy metal concentration in the soil and plant tissues or photosynthetic efficiency, respectively. Undoubtedly, it could improve the quality of the manuscript).
  • Thank you very much. Any stamens not concerned heavy metals was removed

  • To sum up, the discussion should be significantly remodeled in respect to the aim of undertaken study and to show multidimensional and interrelated scientific truth resulting from conducted research. I also encourage the Authors to learn more about oxidative stress and their impact not only on chloroplasts.

  • Thank you very much. The authors tried to improve the discussion. We hope to cover the objective and the results of the work.
  • Conclusions are written exaggeratedly and they are not supported by the results obtained. Moreover, they contradict each other when the statement that ‘the application of Ag-nanoparticles and  ginsenosidiumtans increased adverse mercury effects on the Trigonella foenumgracum plant by reducing vegetative growth RWC and chlorophyll contents’ is taken into account together with the following one about AgNPs and S.g. usefulness ‘to energize the growth and yield in stressed plants’.
  • Thank you very much. The authors tried to rephrased and considered the results obtained.
  • The increased level of H2O2indicated oxidative stress and it is not absolutely connected with stress mitigation! An enhanced accumulation of mentioned non-enzymatic antioxidants together with H2O2 content seems to confirm this hypothesis. It cannot be also concluded about mercury uptake since its content was not determined. Undoubtedly, Ag-NPs or/and symbiotic microorganisms may help to combat stress conditions, but it was not observed in presented results.
  • Thank you very much. You are entirely correct, but our work may not fully cover the HM and Nanoparticls levels adequately, and further studies are needed.

Some detailed comments:

Key words: delate thyme; phenolic compounds would be better: Thank you very much. The suggestion is considered

Statements in lines 79-80, 102-104, 119-120, etc. are not the results. Please, avoid any general comments in the result section and focus only on obtained data.

Thank you very much. The suggestion is considered.

Line 104: these features were monitored not only at higher concentration. By the way, higher than what? Thank you very much. The statement is corrected

The abbreviation ‘ns’ does not occur in the tables. Thank you very much. The abbreviation deleted

Line 117: what does the ‘second level’ mean? Thank you very much. It is clarified.

Line 139: 6 instead of six :

Thank you very much. The figure is corrected

Please, remove brackets when NPs and S.g. combination is mentioned in every tables.

Thank you very much. The brackets are removed.

Line 359: provide full name and explain how was this species identified and obtained.

Thank you very much. Done

How was the HgCl2 applied? Once? How were the plants watered during cultivation? How was the leaching of Hg from the soil stopped?

Thank you very much. In the materials and method section, treatments are clarified

Line 411: change the order on H2O2, proline, TPC (it refers also to the description)

Thank you very much. Done.

Line 426: Reference [74] is redundant; by the way, it does not exist in the reference list.

Thank you very much.  The reference is added

Anthocyanins are not the photosyntetic pigments! They represent phenolic compounds and the procedure for their extraction/determination should be described together with TPC.

Thank you very much.  Corrected

Specify for which analysis the ninhydrin was used. Explain how the proline and TPC levels were estimated (using calibration curve?) Equivalent of which rea`gents were used for quantification?. Provide more detailed information about concentrations of used reagents for H2O2 determination.  

Thank you very much. In the materials and method section, treatments are clarified

The unit for H2O2 is unacceptable and values must be recalculated in a proper way for this parameter in order to show the H2O2 content per gram (not l) of tissue.

Thank you very much. The unit is corrected   

There are numerous typing and language errors that could be carefully removed from the whole text.

Thank you very much. The authors thoroughly tried to correct typing and language errors

Reviewer 2 Report

Manuscript can be accepted in present form.

Author Response

Dear professor

Thank you very much for your effort that improved our manuscript. 

Best regards 

Round 2

Reviewer 1 Report

Dear Authors,

although your responses to my comments are quite limited and do not provide any explanation, I believe the changes in the manuscript are reliable. There are still some typing errors in the revised text (mostly in tables), such as different font or no subscript, but in this case errors will be possible to be eliminated during further editorial processing. Nevertheless, I have a few more minor comments: 

- change in the whole text 'Sd' (standard deviation) on 'SD'

- lines 115, 407: carothenoids are photosynthetic pigments and therefore they shoould not be distinguished!

- line 132, 414: it should be 'anthocyanins' not 'anthocyjanine' - please check this in the whole text

- line 134: put the shortcut at the end, after contents; put uppercase also for phenol and contents since it is TPC (not Tpc). By the way, TPC is commonly reserved for 'total phenolic contents' not for 'phenol content' and thus it need to be improved in the whole manuscript. It is particullarly important because the Authors did not measure phenol concentration as the simplest phenolic compound but the general content of many other phenol-based derivatives, such as eg. gallic acid used here as a standard for TPC estimation.  

- line 200: delete 'potency'

- line 230: provide full name for E. coli

- line 411: as I asked before, please change the order. Firstly it should be mentioned about stress molecule, i.e. H2O2, then defense components, i.e. proline, TPC. It refers also to the description in this paragraph

-line 426: reference [74] is still rebundant, since it does not concern literature, scientific articles, etc, but it refers only to software

- line 427: Vis instead of vis

Author Response

Dear professor

Thank you very much for the comments and suggestions raised that help to improve our work. Please find attached the file of the response. All corrections are inserted as track changes.

Best regards

This manuscript is a resubmission of an earlier submission. The following is a list of the peer review reports and author responses from that submission.

Round 1

Reviewer 1 Report

Thank you very much for inviting me as a reviewer for manuscript entitle "Morphological and physiological responses of mercury-stressed Trigonella foenum-gracum L. towards biosynthesized silver nanoparticles and Sphingobacterium ginsenosidiumtans". However, for the quality of the journal, I cannot accept this manuscript for publication. Below, you will find the explanation for my decision.  

The title of the manuscript is inadequate to the content. The authors do not show the effect of Ag-NPs and Sphingobacterium ginsenosidiumtans, but only the effect of mercury application on plants, which is nothing new. What is more, the objectives for the research are unclear, and the research hypothesis are not indicated. However, the main objection is about result presentation and interpretation. This may be the result of a wrong research goal set by Authors. The values obtained from individual treatments cannot be averaged and then statistically tested. For this reason, the calculation should be made once again to show how Ag-NPs, S. ginsenosidiumtans or their combination influence on plants within particular Hg treatments. By the way, using two-way analysis of variance requires the specification of both factors and their effects on evaluated variables. According to the present paper the application of tested agents makes no sense and does not help to combat metal presence, but rather contributes to oxidative stress. Consequently, H2O2 content as well as non-enzymatic antioxidants increased resulting in plant growth disturbances. Thus, conclusions are written exaggeratedly and the discussion section is very speculative. Moreover, units for pigments are not shown, and taking into account differences in tissue water content, data should be expressed per DW not FW.

The text is also full of incorrect statements and the nomenclature used, e.g. chlorophyll membrane; anthocyanins presented as photosynthetic pigments or heavy metals that may be transformed into more water-soluble form or even released into the environment by phytovolatilization (in the case of plants, this process concerns only Se, not heavy metals in general). Finally, there are some language errors, including grammatical, syntactical, typing, punctuation, noun plurals, verbs and other weaknesses.

Reviewer 2 Report

This paper approach an interesting topic. Experimental data and results are clearly presented therefore is worth to be published in your journal

Reviewer 3 Report

The manuscript entitled “Morphological and physiological responses of mercury-stressed Trigonella foenum-gracum L. towards biosynthesized silver nanoparticles and Sphingobacterium ginsenosidiumtans is good but the following concerns should clarify before further process. 

  1. In the abstract, line no 28, S. ginsenosidiumtans should be italic.
  2. In Table 1, line no 108, sentence starting from ‘Mean±Sd values for treatment over three replications. According to Tukey's test, different letters within the same columns show significant differences (p < 0.05). The higher cases are the differences among HgCl2 treatments, and the lower cases are the differences among AgNPs and S. g treatments mean. ns: means not significant’ should be below the table.
  3. In table 2, line no128, similar changes are needed.
  4. In table 2, Chlorophyll spelling should be correct.
  5. Similarly in table no 3.
  6. In Table no 3, the table should be rearranged, in the first treatment, it should be ‘control’ as it is showing ‘trol’.
  7. Authors should be made interaction between the concentration of HgCl2 and different treatments of AgNO3 and S. ginsenosidiumtans with control. As in Table 1, I don’t understand why the authors didn’t use statistics between treatments of a particular concentration of HgCl2? 
  8. Why authors did Tukey test for analysis in this experiment? Why not DMRT?
  9. In line no 396, no space should be there in the formula.
  10. Kindly rechecked throughout the manuscript and use the correct superscript or subscript for the abbreviated name as in the conclusion (line no 426) H2O2 should be H2O2.
  11. In the reference section, Ref 10, line no479, ‘Plant Probiotic Bacteria’ should be ‘plant probiotic bacteria’.
  12. In the reference section, Ref 19, line no501, Journal name abbreviation need dot (.), or year should be bold.
  13. Do not begin sentences by abbreviation/symbols etc.
  14. I recommend authors maybe a little more information “Role NPs mitigate the pernicious outcomes of heavy metal stress in crop plants. A recent report (DOI:10.1016/j.enmm.2021.100457) might be helpful.